# Combined Unplugged and Educational Robotics Training to Promote Computational Thinking and Cognitive Abilities in Preschoolers



Chiara Montuori [1,*], Gabriele Pozzan [2], Costanza Padova [1], Lucia Ronconi [1], Tullio Vardanega [2] and Barbara Arfé [1]

1   Department of Developmental Psychology and Socialization, University of Padua, 35131 Padua, Italy; dipartimento.dpss@unipd.it (C.P.)
2   Department of Mathematics, University of Padua, 35131 Padua, Italy; gabriele.pozzan@phd.unipd.it (G.P.)
\*   Correspondence: chiara.montuori@phd.unipd.it

**Abstract:** Computational thinking (CT) learning activities are increasingly integrated in early-stage school curricula in several countries. Tools used to teach CT in early school years include unplugged coding—i.e., programming without computing devices—and educational robotics (ER)—i.e., giving instructions to a digitally controlled mechanical robot to perform specific actions in a physical environment. Past studies have shown that training coding skills through ER enhances first graders' executive functions (EFs). Little is known, however, about the effects of ER interventions, alone or combined with un-plugged activities, on preschoolers' CT and EF skills. In a cluster-randomized controlled trial, we assessed whether improvements in preschoolers' coding skills, following interventions based on combinations of unplugged coding and ER, transfer to plugged (computer-based) coding abilities and to EFs such as planning, response inhibition, and visuo-spatial skills. Forty-seven preschoolers from four class groups, with no prior exposure to coding, were randomly assigned to an experimental (unplugged coding and ER, two classes) or control (standard school activities, two classes) instructional groups. Four coding tasks, one standardized planning task (Tower of London test), one standardized response inhibition task (NEPSY-II inhibition subtest), and one visuo-spatial standardized task (Primary Mental Ability subtest) were used to assess children's skills at the pretest (before the intervention) and posttest (after the intervention). To measure retention, the same skills were also assessed for 22 children from the experimental group 3 months from the posttest (follow up). The paper discusses the results of this experimental intervention. The results show significant positive effects of the instructional program on children's computer-based coding skills and cognitive abilities, particularly visuo-spatial skills. Between pretest and posttest, children in the experimental group improved in coding, z = 3.84, *p* = 0.000, r = 0.87, and in visuo-spatial skills, z = 3.09, *p* = 0.002, r = 0.69. The waiting list control group showed improvements in coding skills only after the intervention, at the assessment point T3, z = 2.99, *p* = 0.003, r = 0.71. These findings show that practice with tangible and unplugged coding during the last year of preschool not only significantly improves children's skills to solve computer-based coding problems (near-transfer effect), but it may also have some far-transfer effects on cognitive functions, such as visuo-spatial skills.

**Keywords:** preschoolers; educational robotics; unplugged coding; computational thinking; executive functions; visuo-spatial abilities; response inhibition; planning

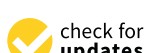



## 1. Introduction

One in three 13-year-old students in Europe currently lacks the basic digital skills and problem-solving skills that are so much needed in modern societies [1]. Computational thinking (CT) learning programs address this need. There is a clear drive in governments in Europe, the USA, and other continents to integrate the teaching of CT in school curricula from early grades. Nonetheless, robust evidence is still lacking on what is the most effective

way to introduce young children, totally novice to coding, to the concepts and processes of computational thinking. Although several studies are emerging in the field, researchers have mainly been concerned with testing the effectiveness of interventions addressed to primary school children or older students (e.g., [2–4]). Computational thinking, which promotes a problem-solving attitude inspired by computer science, encourages students to break down complex problems into smaller, more manageable parts, to recognize recurring problem-and-solution patterns in them, and to develop algorithmic solutions to them that can be executed by an external agent. This aggregate of skills fosters a systematic and organized approach to problem solving [5]. Coding, which stimulates the skills to create, modify, and evaluate program text, fragments, symbols, and the familiarity with programming concepts and procedures, is one of the instruments of CT. Introducing CT from preschool years equips students from the onset of their education with skills and mindsets that are deemed crucial to operating in the modern world, such as digital skills, logical reasoning, problem solving, and creativity [6]. In addition to that, early exposure to CT also aids future learning achievements in several disciplines, particularly Science, Technology, Engineering, and Mathematics (STEM) [7]. In Italy, the teaching of CT from preschool became compulsory with a law issued by the Ministry of Education (motion n. 1-00117, 12 March 2019). However, similarly to what happens in countries within and outside the European Union, the teaching practices of CT in Italian preschool and primary schools are still far from being homogeneous, as many teachers lack the fundamental training and knowledge to achieve this goal. In fact, in Italy CT is currently absent from university curricula for primary school teachers in training.

According to the literature, the tools used to teach coding elements of CT skills during the early school years are: (1) unplugged coding, that is, programming without the use of digital devices; (2) educational robotics (ER), where learners give executable instructions to a programmable robot in order for it to perform specific actions in a physical environment [8]; and (3) plugged coding, which entails developing executable programs to instruct a digital sprite to achieve a goal in a constrained digital environment [2,9–11]. Although these tools are all used to introduce children to coding, they have different characteristics that make them suited for different age groups. Plugged practices make use of computing devices, while unplugged practices do not. Unplugged coding activities involve logic games, cards, strings, or physical movements that are used to represent and understand CT concepts, such as algorithms, without requiring abstract coding [3,12,13]. Although the efficacy of these tools for learning to code has been tested in some studies, very few experimental studies involved preschoolers, and none of them, to the best of our knowledge, tested the effects of combined unplugged and ER interventions. Moreover, none of these studies examined the transfer of ER or unplugged coding skills to children's cognitive abilities, except for problem-solving and visuo-spatial skills.

Whether acquired knowledge and skills can be transferred from one context or problem to another is a key question for educational and cognitive psychologists. In fact, the transfer of learning effects across contexts, tasks, and abilities lies at the very heart of education and in the concept of learning itself, involving the ability to flexibly apply what has been learned [14]. Transfer in learning means that learning in one context impacts learning and performance in other contexts. The transfer of abilities depends on the analogy and overlap between the contexts and problems in which the skills were gained and those presented later [15]. State-of-the-art research suggests distinguishing between near-transfer and far-transfer. Near-transfer is the transfer between the tasks or skills trained and similar or closely related tasks or skills; far-transfer is the transfer between dissimilar tasks or skills, and thus is considered more difficult [16]. In the CT context, near-transfer effects are assessed by examining the effects of coding intervention to various guises of situations that all explicitly require programming skills (i.e., coding abilities). Far-transfer effects are instead assessed by examining transfer to tasks that require skills less closely or directly related to programming, such as cognitive abilities like response inhibition or planning,

which may be used in coding but are assessed through tasks very different to those used during coding interventions [2,17].

Examining these far-transfer effects of coding is particularly important at the transition from preschool to primary school, as the test of far-transfer effects evaluates the general cognitive benefits of coding, that is, the possibility that learning to code may not only be useful to train 21st century digital skills, but also prove effective to foster children's cognitive arsenal. This is even more important during the preschool years, as the preschool period is characterized by greater "neural plasticity" in which the windows for brain development are all wide open and it is easier to lay down neural pathways for new skills [18–20]. Gained knowledge about the effectiveness of coding in this time window would allow age- and cognitive-developmentally appropriate programs for preschoolers to be incorporated into early teaching curricula with potentially cascading benefits for their learning.

### 1.1. Current Evidence of Near-Transfer and Far-Transfer Effects of Coding during Preschool Years

Several studies conducted with preschoolers have so far demonstrated the feasibility of teaching coding through unplugged activities [13] or ER [21–23]. However, among the studies that have explored the learning of CT skills through ER (see [24,25]), only few have focused on preschoolers [26–29]. Moreover, a recent meta-analysis by Li et al. (2022) highlighted that among the studies focused on the effectiveness of unplugged coding to promote CT skills (e.g., [3,30,31]), none involved preschoolers [32]. Another limitation of the research field is that only a limited number of studies exploring the effects of CT interventions for younger learners are true experimental studies. For instance, just one of the ER studies mentioned above is an experimental study [28]. That study found a positive near-transfer effect of 12 h of ER activities on the algorithmic skills of 42 preschoolers aged 5–6 years, with a medium to large effect size ($d = 0.77$). The sequencing abilities of the participants were however not significantly influenced by the ER intervention [28]. Another experimental study demonstrated positive effects of unplugged coding on CT abilities [31]. However, the intervention addressed older participants (6–8 years).

Although these studies suggest positive effects of instructional interventions based on ER on preschoolers' coding abilities, their focus is on near-transfer effects only, that is, on the effects of the interventions on coding itself. Additionally, the quantity of studies is still too small to inform truly evidence-based instructional practice.

Other experimental studies (e.g., [2]) have demonstrated that learning CT in educational settings through plugged, computer-based, coding activities since grade 1 also improves children's executive functions (EFs), thus showing the far-transfer effects of plugged coding.

EF is an umbrella term that refers to a complex set of cognitive skills related to goal setting and the performance of goal-directed behaviors, which also underpin learning and academic performance across several domains [33–35]. Welsh and Pennington (1988) defined EF as the ability to maintain an appropriate problem-solving set for achieving a goal. This capacity encompasses core skills such as children's command over impulsive responses (i.e., inhibitory control), their ability to update their working memory (WM), and their skill to switch perspectives and shift attention between mental sets or tasks (i.e., cognitive flexibility) [36–39]. These three cognitive functions (i.e., response inhibition, working memory, and shifting) are defined core EFs [37,40] from which higher-order executive functions, such as planning and problem solving, originate [36,41–43].

The preschool period that accompanies the transition to primary school is particularly sensitive for the development of EFs [44,45]. Response inhibition skills develop rapidly between preschool and early school years. Inhibition is already present in the first year of life; it undergoes, however, a significant improvement in the preschool period [46], from the age of 3 to 6 years [43,47,48], and continues to develop throughout childhood and adolescence. Also planning skills seem to develop significantly in the transition between preschool and the first years of schooling [49,50]. It is therefore held that interventions aimed at boosting

the development of EFs are particularly effective during early childhood [51]. Consistently, researchers found that both plugged coding [2,52] and ER [53] significantly affect first graders' EFs (response inhibition, working memory, and planning).

Whereas research on the deployment of coding instruction in preschool, primary, and secondary school is growing rapidly [54,55], the lack of experimental studies testing the effectiveness of these activities on preschoolers' cognitive development represents a significant gap in the research field. Just three experimental studies have tested the effectiveness of ER on preschoolers' cognitive development, with a focus on problem-solving skills [56,57] and inhibitory control task [58]. The results showed that ER enhanced children's problem-solving skills, assessed through an ad hoc problem-solving skills scale [56] and mathematical problem-solving tasks [57]. Conversely, ER did not prove more effective than playful control activities in boosting preschoolers' self-regulation (i.e., performance in an inhibitory control task) [58]. In Yang et al. [58], the (active) control group was exposed to block play activities that also engaged inhibitory control skills by having children strategically determine which building wooden blocks to use to construct a tower. The findings of another recent study [59] are, however, in contrast with these results. The study [59] showed significant positive far-transfer effects on preschoolers' inhibitory control and problem-solving skills of a training intervention that combined unplugged coding, ER, and plugged coding. Inhibitory control skills were assessed through the same self-regulation test used in Yang et colleagues' [58] study (i.e., the Head-to-Toes Task, [60]). Although the aims of this study [59] are innovative as it tested the effects of *combined* training intervention, several experimental details are missing (modality and duration of each intervention), whose omission makes the instructional design not replicable. It is also unclear whether the plugged (based on Scratch) coding activities that were carried out in that study contributed to the efficacy of the training. Arguably, virtual plugged coding requires more abstraction skills and thus could be less appropriate for younger children than unplugged and ER activities, which favor a more concrete programming experience. Unplugged and ER can therefore be especially suitable tangible tools for introducing preschoolers to coding.

Besides the above-mentioned effects on children's EFs, ER and coding interventions also seem to have significant effects on children's visuo-spatial skills. Visuo-spatial skills, which involve the mental manipulation and understanding of visual stimuli in relation to objects and their positions, are involved in visual coding tasks. Unsurprisingly, prior studies have highlighted the association between visuo-spatial skills and programming performance (e.g., [61]). Spatial skills may contribute to the comprehension of complex code structures, visualization of program execution, and debugging, all of which are crucial aspects of successful education in programming. At the same time, since coding involves visuo-spatial skills, it is also possible that performing coding activities, especially unplugged coding, or ER in a physical environment, could improve children's visuo-spatial abilities.

Recent research [62] has also found that coding activities in a tangible environment enhance preschoolers' visuo-spatial skills. The study involved 84 preschoolers aged 66.23 months divided in an experimental group (ER), an active control group (unplugged coding), and a passive control group. Children's visuo-spatial abilities were assessed before and after a series of 10 intervention sessions. Sub-tests from the Test of Visual Perceptual Skills (non-motor) revised (TVPS) battery [63] were used to measure spatial relations and visual memory. Mental rotation skills were assessed by the spatial rotation test [64]. The findings indicated that both unplugged coding and ER improved spatial relations and mental rotation skills, although the effect size was higher in the group exposed to ER (eta-squared = 0.58) than in the unplugged coding (eta-squared = 0.14). No other studies that we know of have tested the effects of ER or unplugged coding on children's visuo-spatial skills.

In summary, although emerging research results show significant effects of ER and unplugged coding on preschoolers' cognitive skills, the studies are still too limited, and most are focused on one or two cognitive skills. During preschool years, both visuo-spatial

abilities and EFs undergo significant developmental changes. Hence, testing the effects of coding across these cognitive skills becomes important to design coding interventions aimed both at teaching CT and training cognitive abilities.

In the present study, we aim to extend the research on the effectiveness of tangible (ER and unplugged) coding programs addressed to preschoolers, testing the effects of a combined ER and unplugged coding intervention on the acquisition of coding skills as well as of three cognitive abilities: response inhibition, planning, and visuo-spatial skills.

### 1.2. The Present Study

In this exploratory study, we conducted a cluster-randomized controlled trial on the effects of a combined intervention involving unplugged coding and ER on preschoolers' coding abilities, and the far-transfer of the coding intervention effects on cognitive EFs and visuo-spatial abilities.

The present exploratory study addressed the following research questions:

1.  Near-transfer effect: Can a combined unplugged coding and ER intervention be effective in teaching 4–5-years-old preschoolers' coding skills and CT processes?
2.  Far-transfer effects on cognitive skills: Do the positive effects of the combined unplugged coding plus ER training transfer to 4–5-years-old preschoolers' response inhibition, planning, and visuo-spatial skills?

## 2. Method

### 2.1. Experimental Design

The study was a cluster-randomized controlled trial (CONSORT guidelines, [65]) with a standard pretest (T1)–intervention–posttest (T2) assessment for the experimental group, and business-as-usual activities for the control waiting list group between T1 (pretest) and T2 (posttest). This latter group received the intervention after T2. At T3, both the waiting list and experimental groups were assessed again. For the waiting list group, T3 was the posttest, in which the effects of exposure to CT were tested against the effects of business-as-usual (standard) instructional school activities. For the experimental group, T3 was a follow-up measure of maintenance of the intervention effects. Four preschool classrooms were randomly assigned to the experimental (coding intervention) or control (business-as-usual) groups. After T1 (pretest), the experimental group, consisting of 2 preschool class groups, practiced coding twice a week for seven weeks. The waiting list group, consisting of 2 preschool class groups from the same schools, performed business-as-usual activities for the same seven weeks, and practiced with coding after T2 (posttest). Such an experimental design also allows the control group to benefit from potentially effective learning activities. For the children in the experimental group, follow-up data were collected at T3, two months after the posttest (see Figure 1).

### 2.2. Participants

Forty-seven 5-year-old preschool children from 4 class groups of the same school in northern Italy participated in the study. All children attended the last year of preschool. None of them had been exposed to coding before the intervention. The four class groups were randomly assigned to either the experimental or the waiting list group.

The experimental/treatment group comprised 25 children (14 girls, 56%; 11 boys, 44%) in 2 class groups, assigned to the treatment group and group participating in coding labs immediately after the pretest (T1). The waiting list group included 22 children (10 girls, 43.5%; 13 boys, 56.5%) in 2 class groups, assigned to business-as-usual activities and receiving the coding intervention only after the posttest (T2). Age ranged from 59 months to 70 months in the experimental/treatment group and from 58 to 70 in the waiting list group. The mean age of the participants was 64.68 months (SD = 2.72) in the experimental group, and 64.82 months (SD = 3.84) in the waiting list group. Children's socio-economic status (SES) was assessed by means of a socio-demographic questionnaire that parents returned along with the written informed consent to participate in the study. SES was

estimated based on parents' education, on a scale from 0 (less than elementary school) to 4 (college), and occupation, from 1 (unemployed) to 4 (professional roles). A composite score was calculated as the sum of the highest education score and the highest occupation score obtained by either parent [2], with a maximum score 8. The mean SES was 7.04 (SD = 1.17) for the experimental group and 7.23 (SD = 0.75) for the waiting list. Familiarity with technology was also gauged by asking parents about children's daily use of digital devices (personal computers, smartphones, or tablets) in their home environment. A composite score was calculated as the sum of the (intensity of) use of those three digital devices. A total of 20 children (80%) in the experimental group were not familiar with any type of mouse-controlling device; 3 children used the traditional handheld mouse (12%); only one child was familiar with the touchpad. In the waiting list group, 20 children (87%) were not familiar with any type of mouse and 2 children were able to use both types (8.7%).

| Experimental Group | | | | |
|---|---|---|---|---|
| **PRETEST** | **CODING** | **POSTTEST** | **STANDARD** | **FOLLOW-UP** |
| Coding + Executive Functions + Visuospatial skills | unplugged coding + ER | Coding + Executive Functions + Visuospatial skills | business as usual activities | Coding + Executive Functions + Visuospatial skills |

| Waiting-list Group | | | | |
|---|---|---|---|---|
| **PRETEST** | **STANDARD** | **POSTTEST** | **CODING** | **POSTTEST** |
| Coding + Executive Functions + Visuospatial skills | business as usual activities | Coding + Executive Functions + Visuospatial skills | unplugged coding + ER | Coding + Executive Functions + Visuospatial skills |
| T1 | 7 weeks | T2 | 7 weeks | T3 |

**Figure 1.** Experimental design.

The treatment and waiting list groups were equivalent for age, t (45) = −0.14, *p* = 0.89, SES, t (45) = 0.64, *p* = 0.52, and familiarity with technology, t (45) = −1.24, *p* = 0.22. A chi-square analysis confirmed that the treatment group and the waiting list group were also homogeneous for gender distribution, χ2 = 0.75, *p* = 0.39. Table 1 shows the demographic characteristics of the sample.

**Table 1.** Demographic characteristics of the sample. Means, standard deviations, and *t*-test (t) of age, socio-economic status (SES), and familiarity with technology (Fam Tech).

| Variable | Group | | |
|---|---|---|---|
| | Waiting-List (N = 22) M (SD) | Experimental (N = 25) M (SD) | *t* (DF) |
| Age (months) | 64.82 (3.84) | 64.68 (2.72) | 0.14 (45) |
| SES | 7.23 (0.75) | 7.04 (1.17) | 0.64 (45) |
| Fam Tech | 1.17 (0.89) | 1.48 (0.82) | −1.24 (46) |

*2.3. Procedure and Materials*

2.3.1. Instructional Design

A combined unplugged and educational robotics intervention was proposed for children to introduce them to coding. The training duration was seven weeks, two 60 min sessions per week, for a total of 14 sessions.

The intervention schedule and organization were discussed with the schoolteachers, who also collaborated in the implementation. To ensure that all children received the same share and dosages of activities and interventions, we first recorded attendance on a logbook, so that, by knowing which children had missed out any activity, we could involve them in recovery sessions. Children's engagement was assessed through observations, and all children were engaged by the researchers as much as possible in the activities, discussing with them coding problems and their solutions.

We decided to plan a combined unplugged coding and ER intervention for several reasons. Firstly, although ER has the merits of being an interactive tool, of a tangible nature and therefore particularly suited to the sense-motor experience typical of pre-school age, ER is also a complex tool for younger children, especially for those without prior experience of coding. In addition to the learning demands of children coding with no knowledge of programmable robots, children need to acquire a certain number of instrumental and command skills to manage the robot. All these stimuli make demands on younger children's memory and cognitive system, which induce children to focus on those demands rather than on programming concepts and processes. Thus, we adopted a more gradual approach to introduce coding to preschoolers, starting with propaedeutic activities and unplugged coding to introduce fundamental CT concepts first (e.g., sequences and algorithms) without the additional demands of learning to use new digital tools. Next, we introduced robots and robot programming as an extension and generalization of the unplugged coding experience. We conjectured that this would have allowed children to consolidate the new concepts gradually before experiencing programmable robots. An additional problem of using robotics with young children is that although perspective taking is crucial for ER as children need to take the robot's perspective (i.e., visual perspective and spatial perspective) to understand which instructions to give it to achieve the given goal, for younger children, taking others' perspectives can be cognitively difficult; unfortunately ER or interacting with robots in general do not seem to help in the understanding of directions from the robot's point of view, as confirmed by a recent study [66]. The authors of that study argue that although the activities with the programmable robot (i.e., *Cubetto*) were engaging and challenging for children, perspective taking proved complex for the preschoolers. Indeed, several facilitations were needed to support children in the task (e.g., giving instructions while sitting behind the robot, moving the mat with all the materials around to ensure that the child kept facing robot's tail, putting red and yellow stickers on each child's right and left hand, respectively, since many 3–6-year-olds cannot yet tell right from left, [66]). To date, just one recent study tested the feasibility of both unplugged coding and ER intervention to teach CT skills to preschoolers [23]. This trial first introduced unplugged coding activities and then the programmable robots. The ER intervention consisted of programming *Beebot*, a bee-shaped robot that is programmed with directional commands via input pressing buttons placed on its back. The robot executes the instructions when the child user presses the "play" button. The authors found that programming *Beebots* was not as easy as expected for young children, who incurred difficulties in using the map grids required to create algorithms to program the robot. In addition to the cognitive difficulties of their use, educational robots are also expensive and not affordable for all schools and teachers. Thus, some authors [23] have stressed the importance of introducing children to CT through unplugged coding activities. Unplugged coding has some advantages over ER and could be a first step to introducing young children to coding, allowing children to acquire such basic skills as distinguishing right/left and experiencing perspective taking in a natural, tangible environment. By combining unplugged and ER (educational robotics) activities within the same instructional program, our goal was to take advantage of the strengths offered by both of these tools.

To maximize children's engagement, we developed a scenario and narrative setting for the training sessions: the story of a robot, named *Cubetto*, who was navigating through space, attempting to reach Earth and, more specifically, to the school where the training took place. *Cubetto* was being aided by a group of bees, i.e., the *BeeBot* robots the children

would play with during the training. Thus, each training session started with a message from the bees, in the form of a letter found in the school's letterbox: the message explained the session's activities and tied them to the overarching story. From a pedagogical point of view, we built the training sessions around three phases:

- Preparatory: activities aimed at ensuring that all children shared a common set of basic skills such as being able to distinguish left and right, basic pattern recognition, etc.;
- Unplugged coding: activities that introduced the concept of code as a precise sequence of instructions executable by mechanical agents that, at this step, were embodied by the children themselves;
- Educational robotics: activities that introduced the *BeeBot* and *Cubetto* robots and consisted of activities during which children could program an actual fully mechanical computing agent.

The following Sections describe the activities of each session in detail. For a summary of the whole training, see Table 2.

**Table 2.** Lesson plans.

| Coding Sessions | Macro -Step | Activities |
| --- | --- | --- |
| Session 1 | Preparatory | Games aimed at developing basic directional skills such as right-left discrimination. |
| Sessions 2–3 | Preparatory | Reproducing "color-codes", i.e., sequences of colors, with construction bricks or colored dots on paper grids. Reproducing sequences of pictorial symbols. |
| Sessions 4–5 | Unplugged coding | Understanding sequences of coded instructions and executing them to create pixel art. |
| Sessions 6–7 | Unplugged coding | Solving navigational tasks on a child-sized map by taking the roles of programmer, robot, and tracer of execution. |
| Session 8 | Unplugged coding | Solving navigational tasks on two-dimensional maps by programming sequences of instructions and executing them by moving a pawn. |
| Session 9 | Educational robotics | Familiarization with the *BeeBot* robots. |
| Session 10 | Educational robotics | Reading, understanding, and mentally simulating pre-written sequences of instructions for the *BeeBot*. |
| Sessions 11–12 | Educational robotics | Programming the *BeeBot* to solve navigational tasks. |
| Sessions 13–14 | Educational robotics | Programming *Cubetto* to solve navigational tasks while inventing stories to justify its travels. |
| Closing session | Educational robotics | Metacognitive reflection on the goals of CT and the meaning of programming. |

Session 1

- Goal: ensuring that all children were able to distinguish left-right directions.
- Narrative: an audio message from the bee characters directed the children to the school's letterbox, where they found the materials for the session.
- Activities: the first activity consisted of a game based on right-left discrimination exercises (e.g., instructors showed left hand, right foot, and the children were asked to move the corresponding body part and name the direction). In the second activity, the children were presented with sheets of papers representing two windows and were asked to open the left/right window to find a hidden figure underneath. The third activity was a game in which the instructors asked the children, one at a time, to pick a fruit randomly placed on a table and place it in a basket set either to the left or the right of the table. At the end of this session, the instructors gave all children a bracelet to put on their left arm to reinforce and aid the recognition of directions during the next sessions.

Sessions 2–3

- Goal: pattern recognition and reproduction of a sequence of instructions.
- Narrative: another vocal message from the bees introduced the concept of code as a sequence that must be executed, in this case reproduced, exactly, step-by-step. The

message also introduced the idea of debugging by stressing that errors in coding should not be discouraging and that they may and should be corrected.

- Activities: during the first activity, the children received sheets of paper representing sequences of colors and were asked to reproduce them with toy construction bricks. The bricks were placed in baskets positioned at a certain distance from the table the children were working on, so they had to carefully observe the paper sequence and remember the number of bricks they needed and their colors. Children who committed a mistake in retrieving the bricks would recognize it while building the sequence and could then go back to the basket to fix it. The sequences of colors varied in difficulty starting with just two alternated colors and ending with sequences of all different colors. See Figure 2a for an example of this activity. Similarly, during the second activity, the children were asked to reproduce a "color-code" consisting of a sequence of dots on a 4 × 3 grid drawn on a sheet of paper, with the possibility of having empty tiles (Figure 2b). During the third activity the children reproduced on a 3 × 3 grid the sequence of colors shown on one face of a randomly shuffled Rubik's cube. Finally, the fourth activity introduced codes composed of graphic symbols: the children were asked to memorize a sequence of four symbols—e.g., moon, star, square, circle—and to retrieve them from a different table, without referencing the target sequence. See Figure 3 for an example.

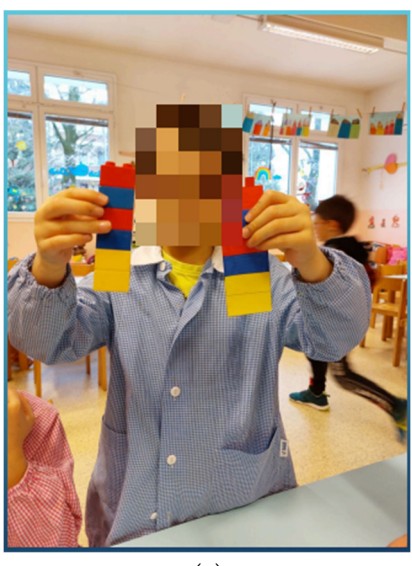 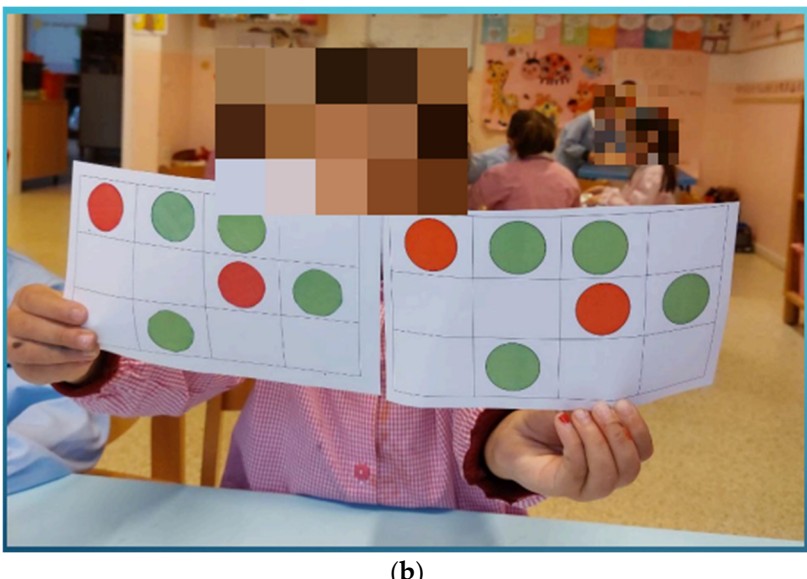

(**a**)                                                                                    (**b**)

**Figure 2.** (**a**) A paper color sequence reproduced by a child with construction bricks; (**b**) reproducing a color code on a grid.

Sessions 4–5

- Goal: understanding sequences of instructions given in a codified form and applying them to create pixel art;
- Narrative: a series of coded instructions was given by the bees to obtain a numerical code which would be used subsequently to access *Cubetto*'s spaceship;
- Activities: during the first activity, the children received (1) large sheets of paper representing a 12 × 12 grid; (2) colored paper tiles to set on the grids; and (3) instructions on how to place the tiles on the grid, given in code form. Figure 4 shows an example of how these instructions were given: each row corresponds to a row of the 12 × 12 grid, the numbers indicate how many tiles of a given color should be placed on the row, the children should "execute" the instructions from top to bottom and from left to right. An important detail is that the instructors did not simply explain how to read the coded instructions. Instead, they encouraged the children to reach the conclusion

by themselves and helped them by asking questions and giving clues e.g., pointing out the fact that the coded instructions had the same number of rows as the paper grids. After having understood how to execute the coded instructions, the children started composing the pixel art on the paper grid. Figure 5 shows an example of this activity. The second activity of this session was very similar to the first one and served to reinforce the notions to be learned. Figure 6 shows the codified instructions (left) and outcome (right) of this activity.

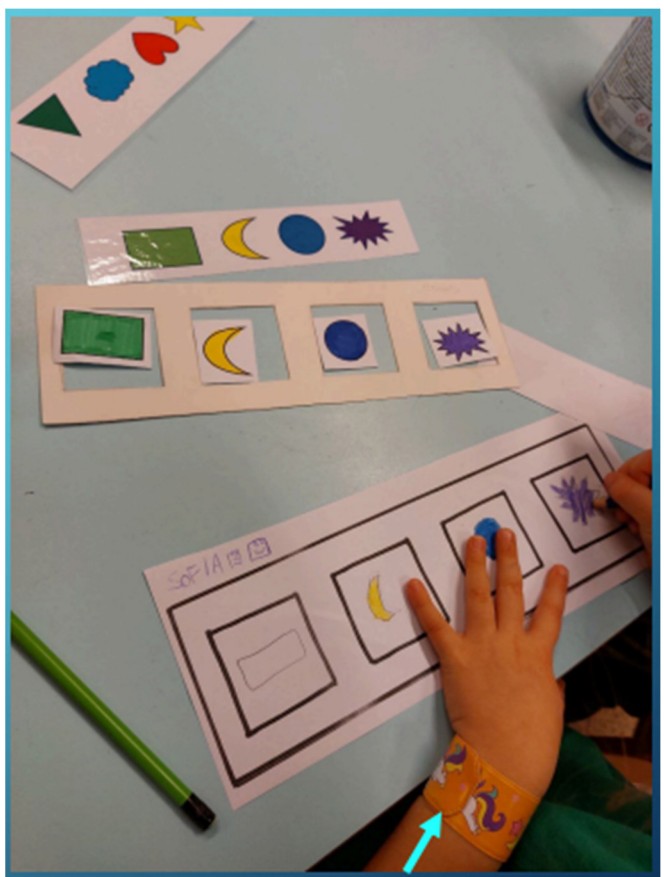

**Figure 3.** Reproducing a sequence of pictorial symbols.

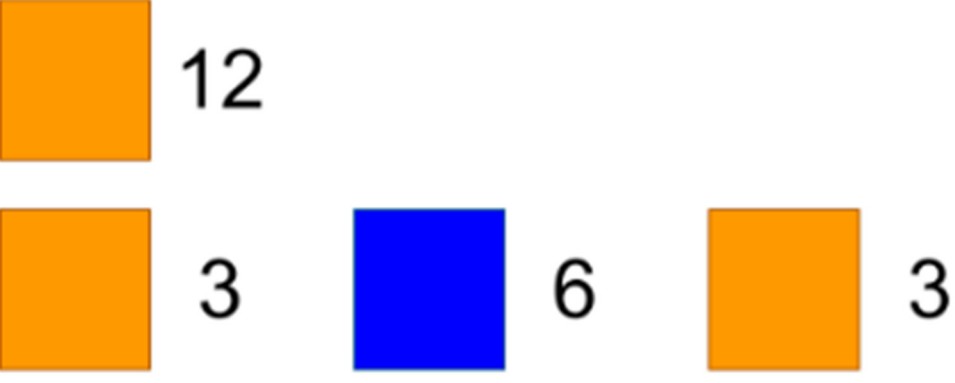

**Figure 4.** Example of coded instructions used to create pixel art.

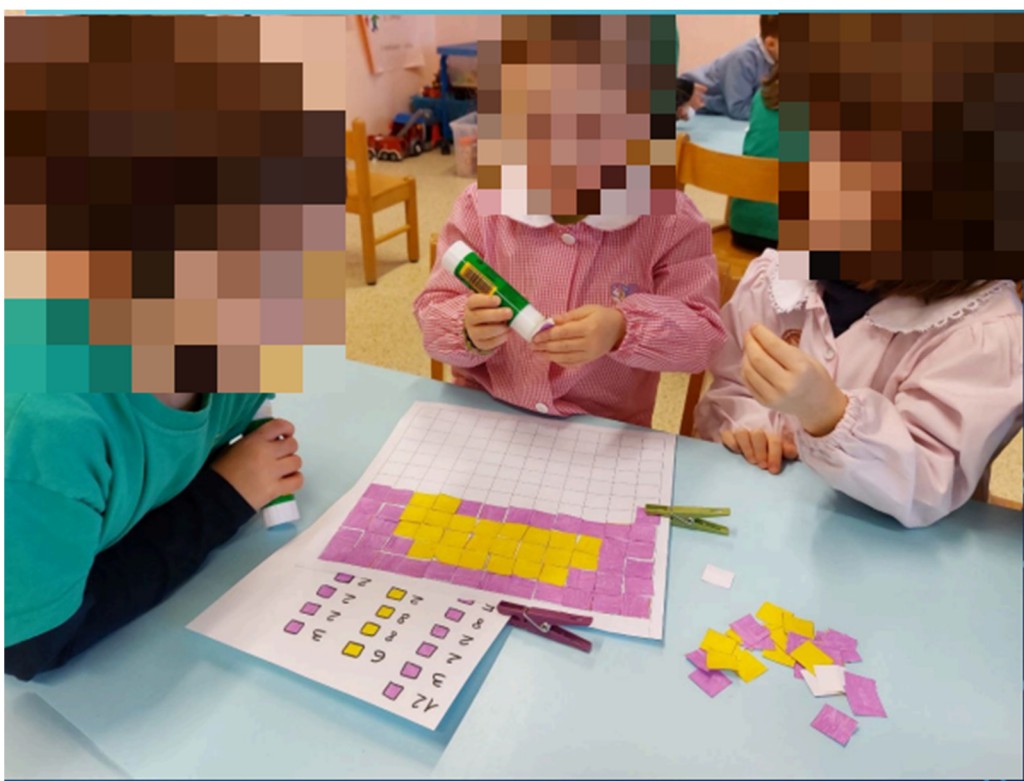

**Figure 5.** Children execute coded instructions to create pixel art.

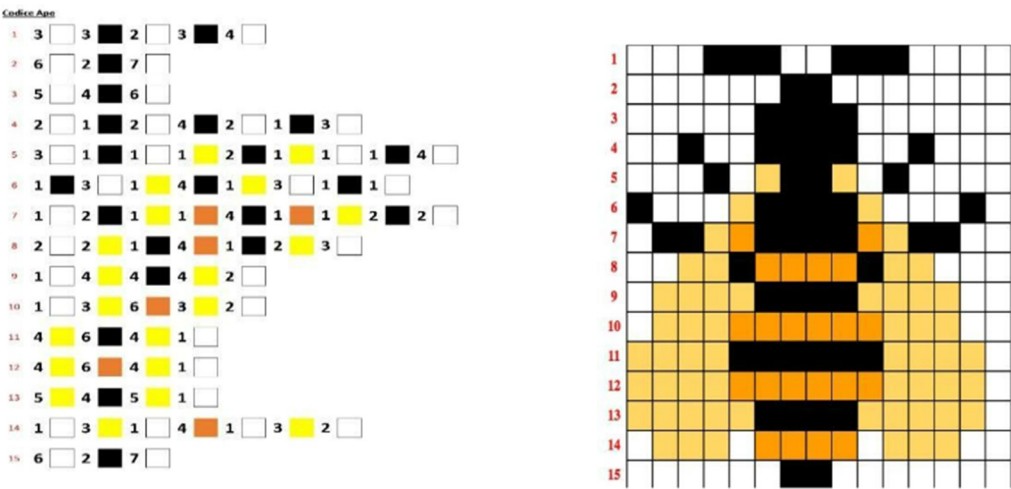

**Figure 6.** Example of a more complex pixel art activity.

Sessions 6–7

- Goal: introduction of the navigational instructions used to program *BeeBot* and *Cubetto* robots and direct experience of the roles of programmer, robot, execution tracer;
- Narrative: the bees tell the children that they will learn to use code to travel around the solar system; their goal will be reaching the sun without hitting any planet;
- Activities: the activities of these sessions took place on a large grid drawn on the floor by using tape. Each tile of the grid was large enough for a child to stand in it comfortably. The instructors used symbols and drawings to mark some of the tiles of the grid: a starting position, a goal position (the sun) and some obstacle tiles (planets). Moreover, the instructors explained to the children the instructions used to interact with this navigational task. These instructions were represented as arrows drawn on

sheets of paper and had a direct correspondence to those used to program *BeeBot* and *Cubetto*: "move forward/backward" one tile according to the direction the robot is facing and "turn left/right" ninety degrees, without changing tile. During the activity, the children took turns in impersonating different roles:

- Programmer: tasked with composing a sequence of instructions to guide the robot from the start position to the goal position, without hitting obstacles (planets);
- Robot: tasked with physically executing the instructions given by the programmer, exactly as told, even if this meant "hitting" an obstacle, i.e., walking on a planet tile;
- Tracer: tasked with marking any tile the robot walked on by placing colored breadcrumbs. These breadcrumbs were used to debug programmers' sequences of instructions. For example, in case of an error, the children could associate each instruction with a breadcrumb and find out which specific instruction caused the robot to hit an obstacle.

Every child was asked to observe the whole process and help capture mistakes. The activities started with very simple tasks, e.g., straight, short paths from the starting position to the goal, and became increasingly more complex by adding more turns and obstacles. See Figure 7 for an example of one of these activities.

Session 8

- Goal: solving navigational tasks by creating sequences of instructions and moving a pawn to execute them;
- Activities: this session moved the navigational games to a smaller scale and tasked the children with solving navigational puzzles represented on sheets of paper by programming a sequence of instructions and then moving a pawn to execute them.

Session 9

- Goal: introduction of the *BeeBot* robots;
- Narrative: this session marked the arrival of the spaceship (Figure 9); the children used the code obtained at the end of Session 4 to open the spaceship, where they finally found the *BeeBot* robots;
- Activities: this session took a more free-form approach and focused on letting the children interact with the *BeeBot* robots and get a basic grasp of their functioning.

Session 10

- Goal: understanding pre-written sequences of instructions and mentally simulating their execution;
- Activities: during this session the children were tasked with reading different sequences of code and then anticipating where they would lead the *BeeBot*, given a starting position and direction on a 6 × 4 tile map. The instructors helped the children by asking relevant questions and pointing out important details such as the starting direction of the robot, the number of instructions in the sequence, etc. For each sequence of instructions, the children placed a marker on the tile they believed the robot would end up in after the execution of the instructions. Then, taking turns, they input the instructions to the robot, watched the execution and verified whether they had given the correct answer. The sequences of instructions started simple and short and grew in complexity and length: 2 steps forward, 2 steps with a turn in between, etc. See Figure 10.

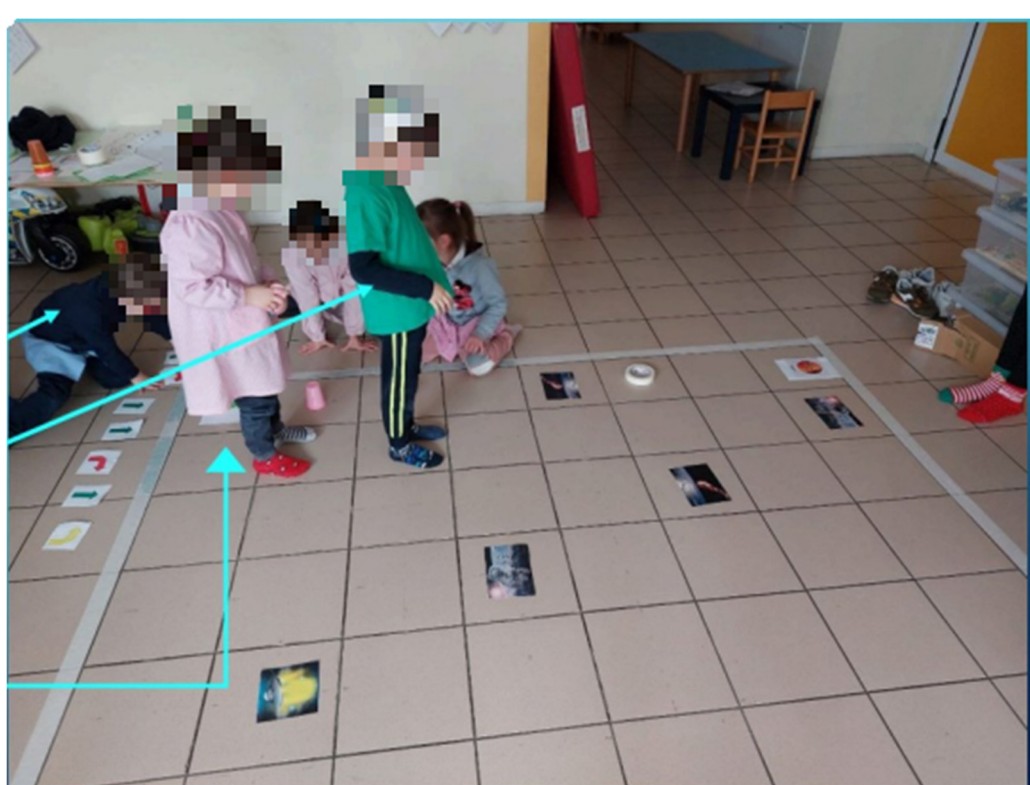

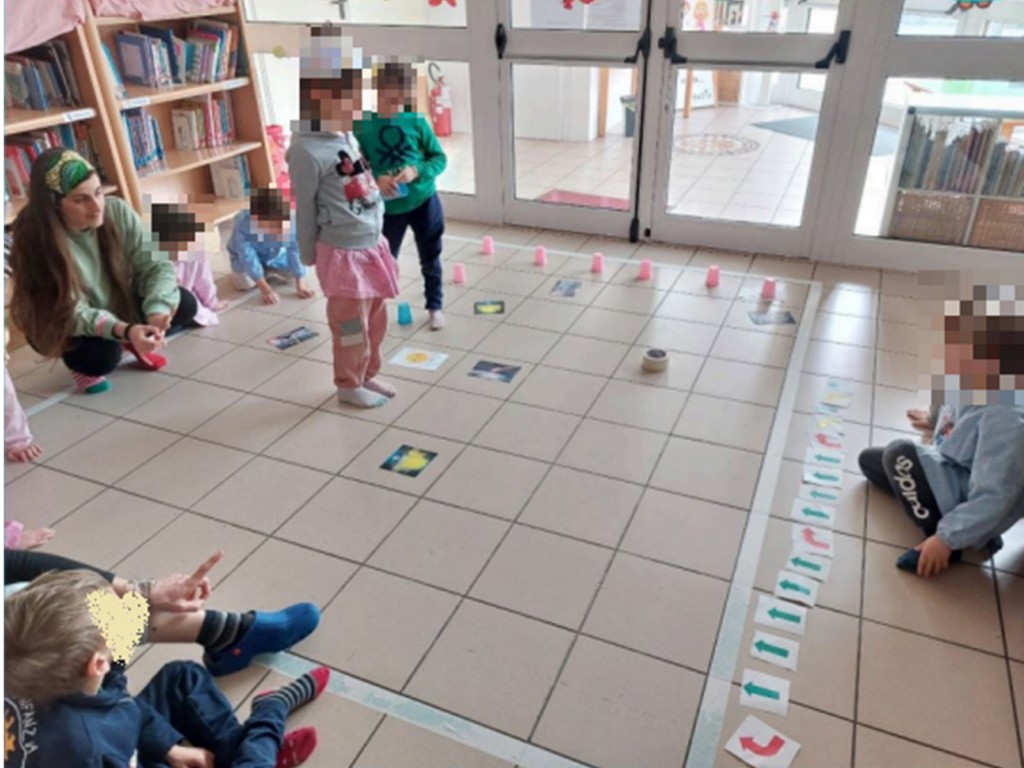

**Figure 7.** Children attempting to solve a navigational task taking the roles of programmer, robot, and tracer.

Figure 8 shows an example of this activity.

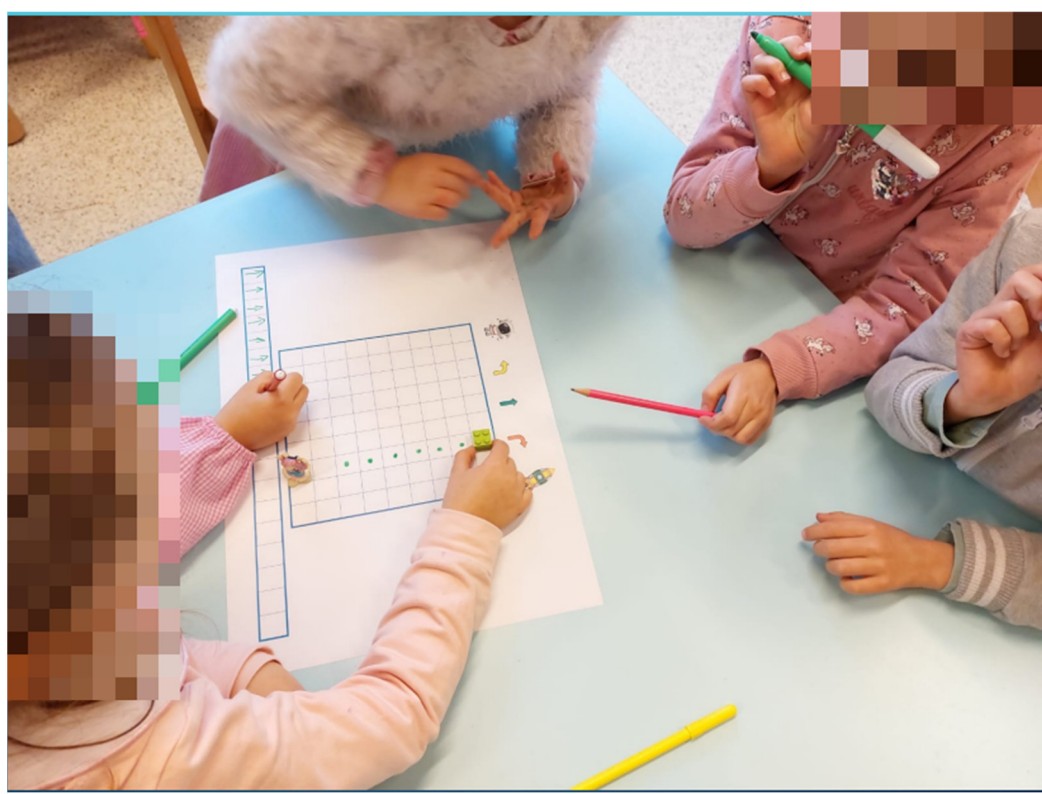

**Figure 8.** Navigational puzzles solved by programming and executing sequences of instructions by moving a pawn on a 2D grid.

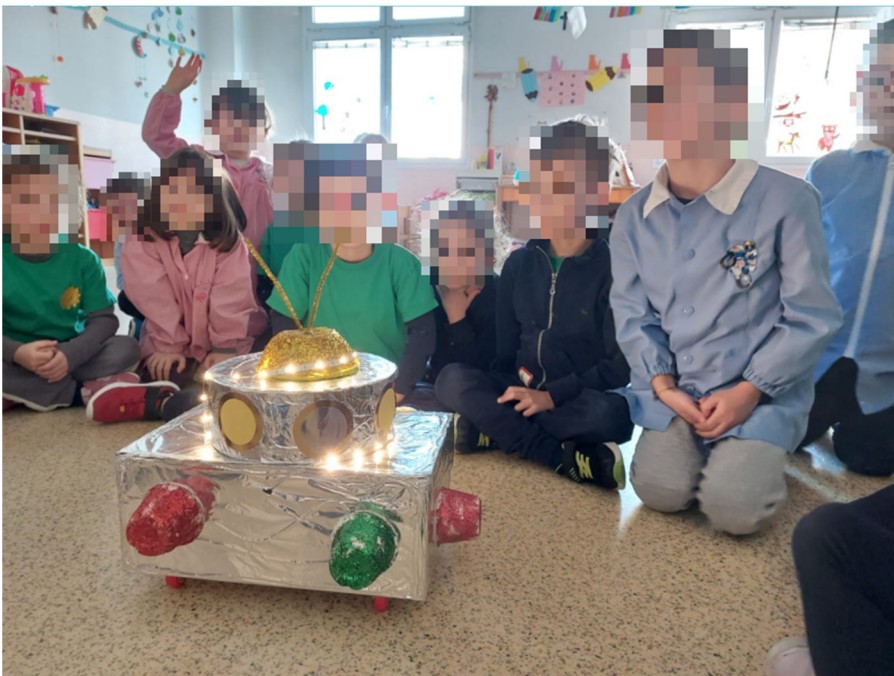

**Figure 9.** The arrival of the spaceship with the padlock that the children opened with the number code obtained during the pixel art session.

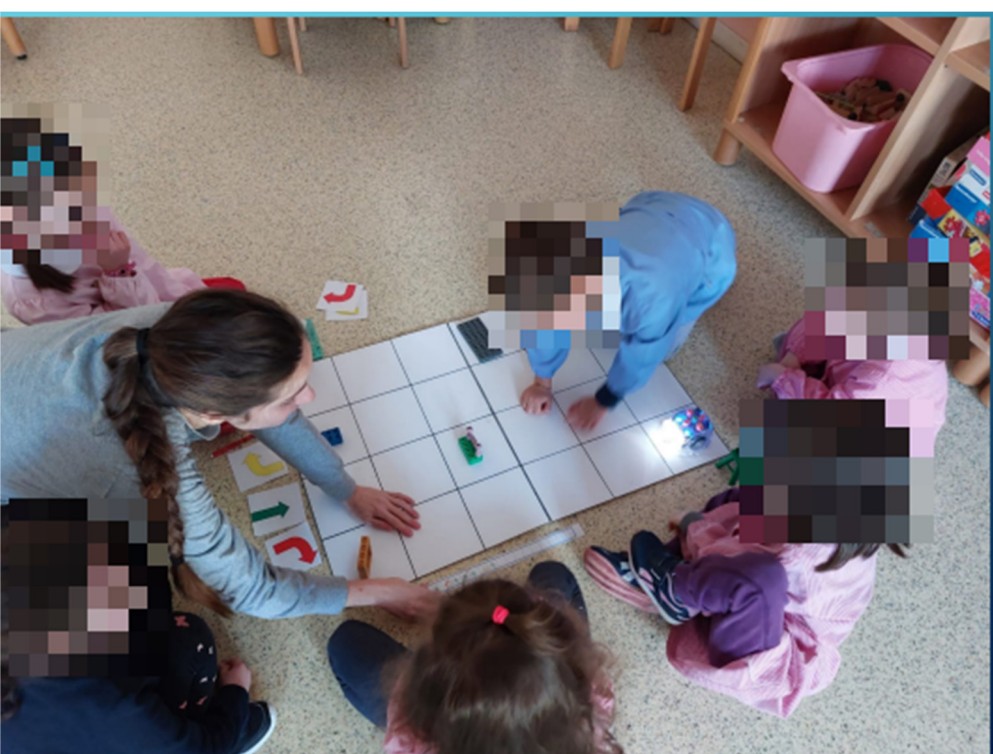

**Figure 10.** Children attempting to understand pre-written sequences of instructions and mentally simulate their execution.

Sessions 11–12

- Goal: programming *BeeBot* to solve navigational puzzles.
- Activities: during these sessions, the children took on different roles while solving navigational tasks that required programming sequences of instructions with the goal of moving *BeeBot* from a starting position to a goal position on a tile-based map. The instructors presented each task by placing a *BeeBot* on a starting tile and a marker on a goal tile (Figure 11). The children took turns in:

  1. Writing a sequence of instructions that would take the robot to the goal tile, given its initial position and direction;
  2. Inputting this sequence of instructions to the *BeeBot*;
  3. Verifying each other's code, e.g., by pointing out possible mistakes.

The tasks were presented in order of increasing difficulty, determined by the length and number of steps of the solutions and the presence of obstacles on the maps.

Sessions 13–14

- Goal: programming *Cubetto* to navigate through a tile-based map representing different situations, and inventing stories to justify its roaming;
- Narrative: these sessions were the culmination of the overarching story of the training activities; the robot *Cubetto* finally reached Earth and the children's school, and the children would now accompany it in its travels. See Figure 12;
- Activities: during these final sessions, the children took turns in solving simple navigational tasks by programming the robot *Cubetto*. To engage all children, even while waiting their turn to interact with the robot, the instructors encouraged them to invent stories to justify the roaming of *Cubetto* on a tile-based map in which each tile represented a particular terrain or destination, e.g., the sea, mountains, desert, etc.

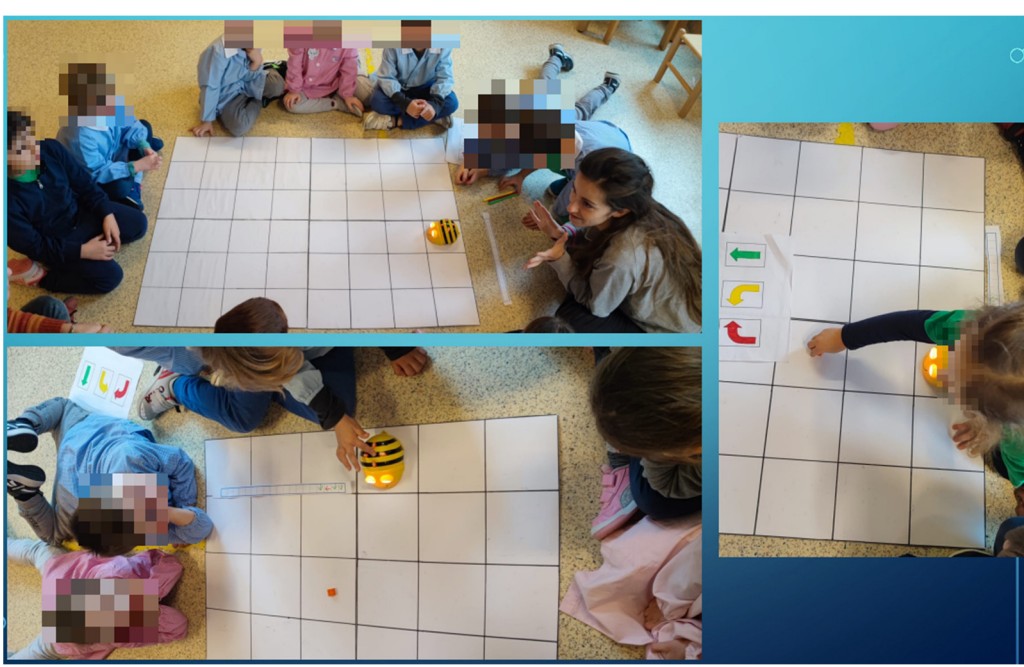

**Figure 11.** Children attempting to program *BeeBot*.

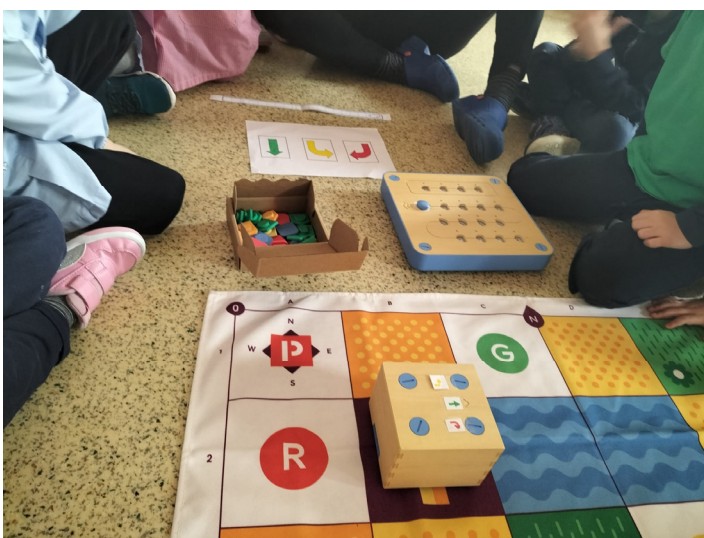

**Figure 12.** Activity with the robot *Cubetto*.

2.3.2. Pretest and Posttest Assessment

At the pretest and posttest, children individually performed 4 coding problems from CoThi platform (see Coding Skills). Moreover, three standardized neurocognitive tests were used, two to assess EFs (response inhibition and planning), and a third one to assess visuo-spatial skills. The use of tasks for executive functions and visuo-spatial skills aimed at ascertaining the far-transfer effects of the intervention on children's cognitive abilities. In the following sections each test is described in detail.

The assessment of coding and cognitive skills was conducted individually for each child. Each participant was supervised by the first author of this study and undergraduate students (trainees and master's students in developmental psychology).

All tasks were administered in a classroom setting, as the school had provided a dedicated space for this research project in which children could participate in the assessment phases without external distractions.

Coding Skills

Before the pretest, both the experimental and the waiting list group familiarized themselves with the CoThi platform and the use of tablets with touchscreen by performing an example trial, assisted by the first author of this paper and trained collaborators (trainee students). The pretest started after this familiarization phase. The coding test consisted of four exercises in which the children were asked to program a sequence of instructions to solve two-dimensional navigational puzzles: guiding a bee sprite through a forest to reach a goal position represented by a star.

The puzzles consisted of tile-based maps composed of path tiles traversable by the sprite and wall tiles creating obstacles. Figure 13a shows an example of such a map: the white tiles represent paths, and the green tiles covered in trees and bushes represent walls. The instructors showed and explained each type of tile to the children before starting the test. To solve the navigational puzzles, the children had to create a sequence of instructions by dragging and dropping code blocks that implement the exact same commands as the *BeeBot*: "move forward/backward" one tile towards the direction the bee is facing, and "turn left/right" ninety degrees on the spot, without changing tile. For example, the puzzle of Figure 13a would be solved by the sequence: "move forward, move forward, turn left, move forward, move forward", shown in Figure 13b. After composing their sequence of instructions, the children would click on a button labelled "Execute" to see whether their coded program solved the problem. The platform executed the sequence of instructions of the children without showing its effects on the map: in case of errors, it would display visual feedback on the screen and the instructors would explain to the child that the program needed some fixing. We decided to avoid showing the execution on the map to (1) reproduce pen and paper unplugged coding digitally and (2) avoid trial and error approaches. The children had up to three attempts to solve each exercise.

We designed the four exercises in a scale of increasing difficulty based on the length of the path and the number and type of turns it contained four exercises of increasing difficulty:

1.  A straight three-tile-long path from the bee to the goal;
2.  A four-tile-long path with a single turn;
3.  A six-tile-long c-shaped path with two turns in the same direction;
4.  A six-tile-long path with three turns in different directions.

This test environment is akin to the unplugged coding and ER activities conducted in the latter training sessions in the sense that it consists of navigational puzzles that must be solved with directional instructions. However, in these tests, the children no longer find themselves physically immersed in the navigational task: they cannot move around the map, change perspective, etc., but must decode and understand the 2D representation they are given and solve it, no longer by pressing physical buttons or placing cardboard arrows on a sequence, but indirectly, by dragging and dropping digital code blocks.

Moreover, not being able to see the actual execution of the program but only receiving feedback stating that there is an error somewhere in the code creates a significantly harder challenge for the children. In particular, to debug their programs, they need to mentally simulate the instructions to find out whether they miscalculated a distance or made a wrong turn.

The CoThi platform recorded the children's coding performance, which was scored for:

*   Coding planning time, in seconds: the time from the moment the child received the visual stimulus (i.e., the map of the exercise) and task instructions to the moment s/he drags and drops the first digital code block;
*   Coding accuracy, in a numerical score: a score of 2 was given if the child successfully solved the item at first attempt; 1 on solving it at the second attempt; and 0 otherwise.

Executive Functions: Response Inhibition

The *NEPSY inhibition (squares/circles) subtest* of *NEPSY-II* [67] was used to assess response inhibition skills at the three assessment times (i.e., pretest, posttest, and delayed

posttest). See Figure 1. Test-retest reliability indices for the age range 5–6 is r = 0.79 for inhibition time and r = 0.77 for inhibition errors [68].

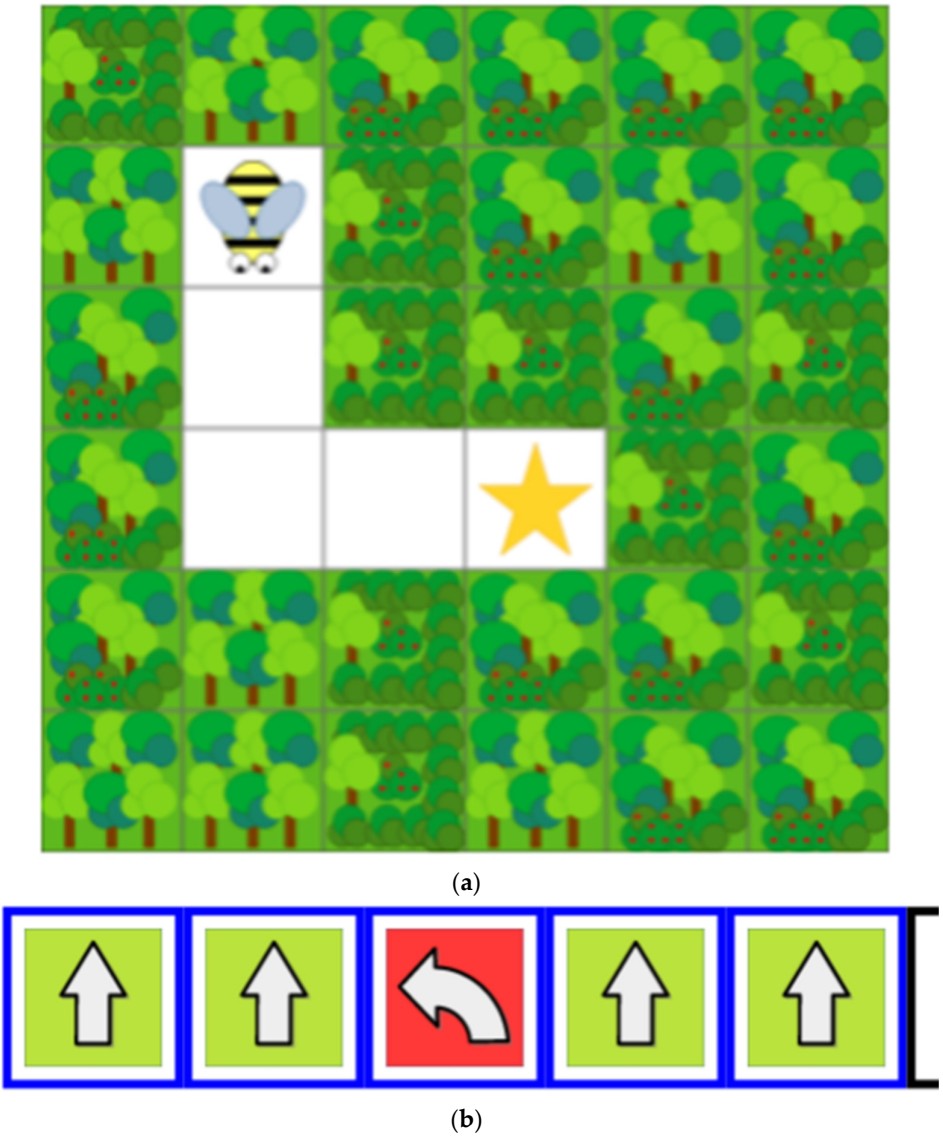

(**a**)

(**b**)

**Figure 13.** (**a**) A 2D tile-based map for a coding test exercise. The goal is guiding the bee to the star, the white tiles are traversable, and the tiles containing trees and bushes are obstacles. (**b**) Sequence of instructions that solve the navigational puzzle.

The *squares/circles inhibition subtest* of *NEPSY-II* consists of naming aloud printed figures (circles and squares) displayed on five rows of eight figures each, saying "circle" for squares and "square" for circles. To respond accurately, children must inhibit their automatic response (i.e., naming "circle" when seeing a circle and vice versa). The task was standardized for children aged 3 to 16. Children's execution time was recorded, and children's performance was scored for:

- Inhibition time, in seconds: the total time to complete the task;
- Inhibition errors: the number of errors and self-corrections made by the child in performing the task.

Executive Functions: Planning

The *Tower of London, ToL*, was used to assess planning [69]. We used a version of the *ToL* standardized for children aged 4–13 [70]. The *ToL* test was used to measure

children's planning and problem-solving skills. The test manual does not report indices of concurrent validity with other measures of EFs. Arfé and colleagues [52] report test-retest reliability indices, i.e., r = 0.57 for accuracy scores and r = 0.71 for planning times. In the same study [52], the concurrent validity of the test was calculated for an age group of 6-year-old children by considering the correlation between the performance on the *ToL* and *Elithorn* tests.

In order to take the test, children interact with a structure composed of a base from which three pegs of different heights rise. The test subjects must move three beads of different colors (red, blue, and green) to reproduce a target configuration; the moves must follow a set of rules:

(1) the child must move only one bead at a time; (2) at any one time at most one bead can be placed on the shorter peg, two on the middle peg, three on the longer peg; (3) the operation must be strictly sequential, i.e., a bead removed from a peg must be inserted on another peg before removing any more beads; (4) a maximum number of moves is allowed on each trial; and (5) each task must be solved within 60 s counted from the moment the examinee receives the visual stimulus of the target configuration. The whole test includes 12 target configurations of increasing difficulty (the number of allowed moves, starting from 2 and arriving at a maximum of 5). Each task starts with the *ToL* in the same initial configuration.

Children's planning skills were scored for:

- Planning time, in seconds, from when the trial is shown to the child until s/he makes the first move, pulling the first ball off the stick;
- Planning accuracy: one point was awarded if the child performed the trial correctly in 1 min without breaking any rule; 0 otherwise.

Visuo-Spatial Skills: Mental Rotation

The *Primary Mental Ability, PMA subtest* was used to assess visuo-spatial skills [71]. We used a version of the *PMA* standardized for children aged 5–17. The *Spatial Relations PMA subtest* used to measure children's visuo-spatial and mental rotation skills was the K-1 level subtest standardized for children aged 5–7. The subtest has good reliability (r = 0.83). The task includes four example trial items and 27 trial items. The items are presented on paper. For each item, a part of a square is presented, and on the right, four similar figures are presented. The child is asked to choose one out of the four figures that would complete the square. Before beginning the test, making sure the child knows what shape a square is essential. Once the child has performed and understood the example trials, the experimenter proposes to continue the activity by performing the next figures. The maximum unfolding time is 6 min.

Child's visuo-spatial skills were scored for:

- Visuo-spatial accuracy: one point was awarded for each trial correctly solved in 6 min; 0 otherwise.

## 3. Results

Seven outliers were identified (with scores in dependent measures >2.5 SD) and deleted from subsequent analyses resulting in a final sample size of 40 (*n* = 22 for the training group and n = 18 for the waiting list group).

As children's scores were not normally distributed, we used non-parametric tests in the data analysis. Between-group differences at the pre-test (T1) were preliminarily explored by a Mann–Whitney U test, which confirmed that the two groups (experimental and waiting list) did not differ significantly in any dependent measure: coding accuracy (U = 167.50, z = −1.03, *p* = 0.30, r = 0.16), coding planning time (U = 189, z = −0.24, *p* = 0.81, r = 0.04), response inhibition errors (U = 127.50, z = −1.95, *p* = 0.05, r = 0.31), response inhibition time (U = 157, z = −1.11, *p* = 0.26, r = 0.18), planning accuracy (U = 179.50, z = −0.54, *p* = 0.59, r = 0.09), planning time (U = 196, z = −0.34, *p* = 0.73, r = 0.05), and visuo-spatial accuracy (U = 174, z = −0.92, *p* = 0.36, r = 0.14).

A Wilcoxon Signed Rank test was used to test the differences within each group between the pretest (T1) and the posttest (T2) when the experimental group was exposed to the coding training and between the posttest (T2) and the delayed posttest (T3), when the waiting list group received the same coding training. In the following, we report the results of the Wilcoxon Signed Rank tests for each dependent measure. The effect sizes (r) estimation was computed in R [72]. Statistical significance was adjusted for multiple comparison to 0.007 (Bonferroni's formula, 0.05/7).

Table 3 reports the groups' mean and standard deviation at pretest, posttest, and delayed posttest, and the statistical comparison between T1 and T2 and T2 and T3.

**Table 3.** Means and standard deviations (SD) at pretest (T1), posttest (T2), and delayed posttest (T3) for each dependent variable, and statistical comparison between T1 and T2 and T2 and T3.

| Variables | Group | Mean (SD) | | z Value | ES (r) |
|---|---|---|---|---|---|
| | | T1 | T2 | | |
| **Coding Accuracy** | Waiting list | 0.61 (0.92) | 0.84 (0.83) | 1.00 | 0.24 |
| | Experimental | 0.36 (0.73) | 3.38 (2.25) | 3.84 * | 0.87 |
| **Coding planning time** | Waiting list | 9.17 (4.47) | 6.46 (2.71) | 2.33 | 0.55 |
| | Experimental | 12.07 (16.18) | 9.04 (5.09) | 0.33 | 0.07 |
| **Inhibition errors** | Waiting list | 1.94 (1.66) | 2.68 (2.26) | 1.53 | 0.35 |
| | Experimental | 3.64 (3.22) | 2.71 (1.98) | 1.04 | 0.23 |
| **Inhibition time** | Waiting list | 52.11 (11.20) | 48.07 (10.44) | 2.42 | 0.57 |
| | Experimental | 57.34 (15.26) | 48.21 (7.10) | 2.45 | 0.54 |
| **Planning accuracy** | Waiting list | 2.17 (1.04) | 3.89 (3.99) | 1.75 | 0.32 |
| | Experimental | 1.95 (1.05) | 3.57 (2.68) | 2.54 | 0.55 |
| **Planning time** | Waiting list | 4.46 (2.24) | 4.24 (1.41) | 0.81 | 0.19 |
| | Experimental | 4.32 (1.16) | 5.06 (2.24) | 2.52 | 0.55 |
| **Visuospatial skills** | Waiting list | 12.21 (3.92) | 14.21 (4.02) | 1.92 | 0.43 |
| | Experimental | 11.09 (3.96) | 15.57 (4.76) | 3.09 * | 0.69 |
| Variables | Group | Mean (SD) | | z value | ES (r) |
| | | T2 | T3 | | |
| **Coding Accuracy** | Waiting list | 0.84 (0.83) | 2.58 (2.32) | 2.99 * | 0.71 |
| | Experimental | 3.38 (2.25) | 2.14 (1.42) | 2.58 | 0.61 |
| **Coding planning time** | Waiting list | 6.46 (3.11) | 6.49 (3.38) | 0.16 | 0.04 |
| | Experimental | 9.04 (5.09) | 5.42 (2.35) | 2.42 | 0.53 |
| **Inhibition errors** | Waiting list | 2.68 (2.26) | 2.37 (1.61) | 0.67 | 0.14 |
| | Experimental | 2.71 (1.98) | 3.18 (3.32) | 0.10 | 0.05 |
| **Inhibition time** | Waiting list | 48.07 (10.44) | 45.26 (9.26) | 1.21 | 0.28 |
| | Experimental | 48.21 (7.10) | 44.13 (10.13) | 1.30 | 0.28 |
| **Planning accuracy** | Waiting list | 3.89 (3.99) | 5.58 (3.91) | 1.89 | 0.36 |
| | Experimental | 3.57 (2.68) | 4.73 (3.43) | 1.07 | 0.40 |
| **Planning time** | Waiting list | 4.24 (1.41) | 4.20 (1.29) | 1.09 | 0.25 |
| | Experimental | 5.06 (2.24) | 4.09 (1.14) | 0.50 | 0.41 |
| **Visuospatial skills** | Waiting list | 14.21 (4.02) | 15.95 (3.81) | 1.95 | 0.45 |
| | Experimental | 15.57 (4.76) | 14.64 (4.26) | 1.07 | 0.24 |

z value = Wilcoxon signed rank test; ES = effect size; * $p < 0.007$, statistical significance was adjusted for multiple comparison to 0.007 (Bonferroni's formula, 0.05/7).

### 3.1. Coding Skills

Between T1 and T2, a statistically significant improvement in coding accuracy was found for the experimental group, $z = 3.84$, $p = 0.000$, with a large effect size ($r = 0.87$), but not for the waiting list group. Planning time on the coding tasks did not differ significantly between T1 and T2 for any of the two groups.

Between T2 and T3, a statistically significant improvement in coding accuracy was found for the waiting list group, $z = 2.99$, $p = 0.003$, after this group received the intervention. The effect size was moderate ($r = 0.71$). Conversely, coding accuracy did not vary significantly between T2 and T3 for the experimental group, which maintained the posttest coding performance. Planning time in coding did not differ significantly between the two times in any of the two groups. The results are represented in boxplots: Figure 14.

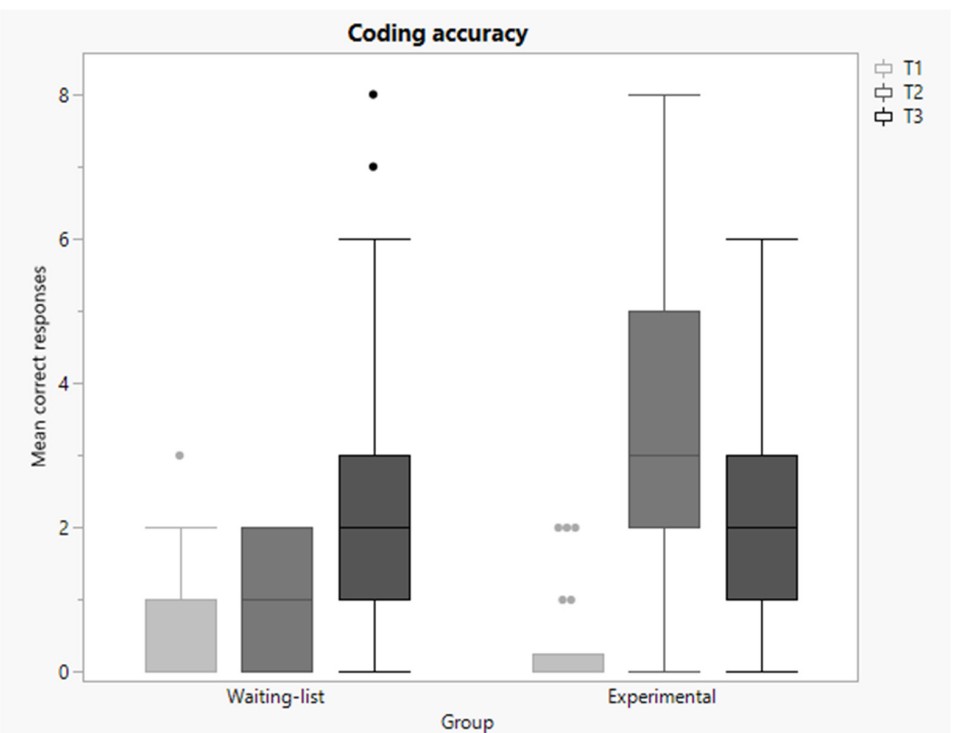

**Figure 14.** Groups' accuracy on coding tasks at the pretest (T1), posttest (T2), and delayed posttest (T3).

### 3.2. Response Inhibition

No statistically significant differences were found between T1 and T2, or between T2 and T3, in response inhibition errors and inhibition time in both groups.

### 3.3. Planning Skills

In the experimental group, differences in planning accuracy between T1 and T2 approached statistical significance, $z = 2.54$, $p = 0.01$, after Bonferroni corrections. The effect size was moderate ($r = 0.55$). For the same group, differences in planning time between T1 and T2 were not statistically significant: $z = 2.52$, $p = 0.02$. For the waiting list group, no statistically significant differences emerged in planning.

No statistically significant differences in planning accuracy and planning time were found between T2 and T3, for any of the two groups. The results are represented in boxplots: Figure 15.

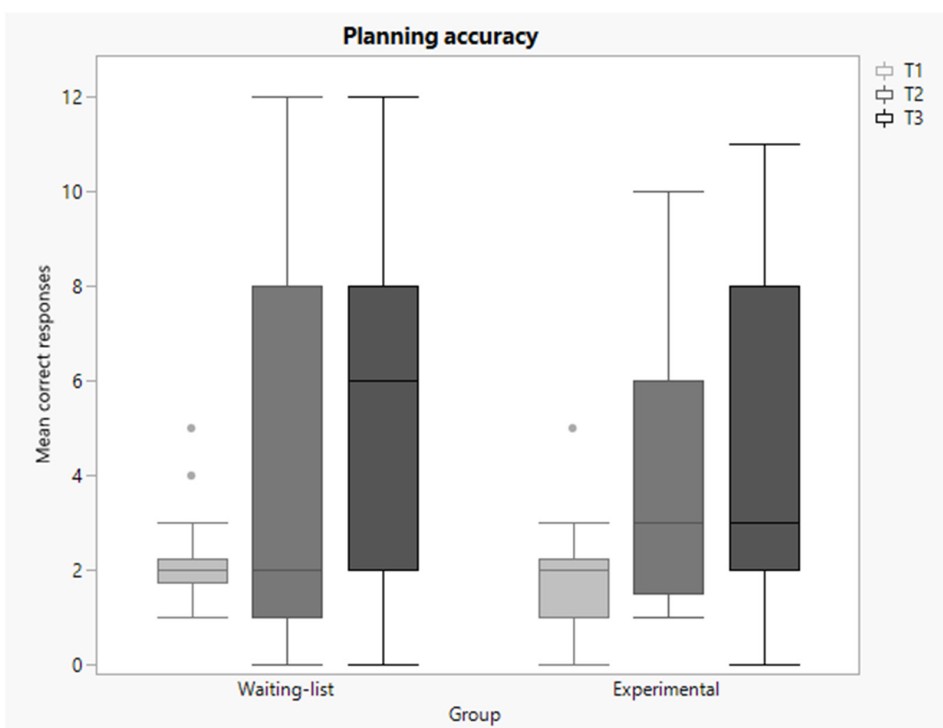

**Figure 15.** Groups' accuracy on planning task at the pretest (T1), posttest (T2), and delayed posttest (T3).

### 3.4. Visuo-Spatial Skills

Between T1 and T2, the experimental group improved significantly in visuo-spatial accuracy: $z = 3.09$, $p = 0.002$. The effect size was moderate ($r = 0.69$). No statistically significant differences were found in visuo-spatial accuracy for the waiting list groups.

Between T2 and T3, none of the two groups showed statistically significant differences in visuo-spatial accuracy. The results are represented in boxplots: Figure 16.

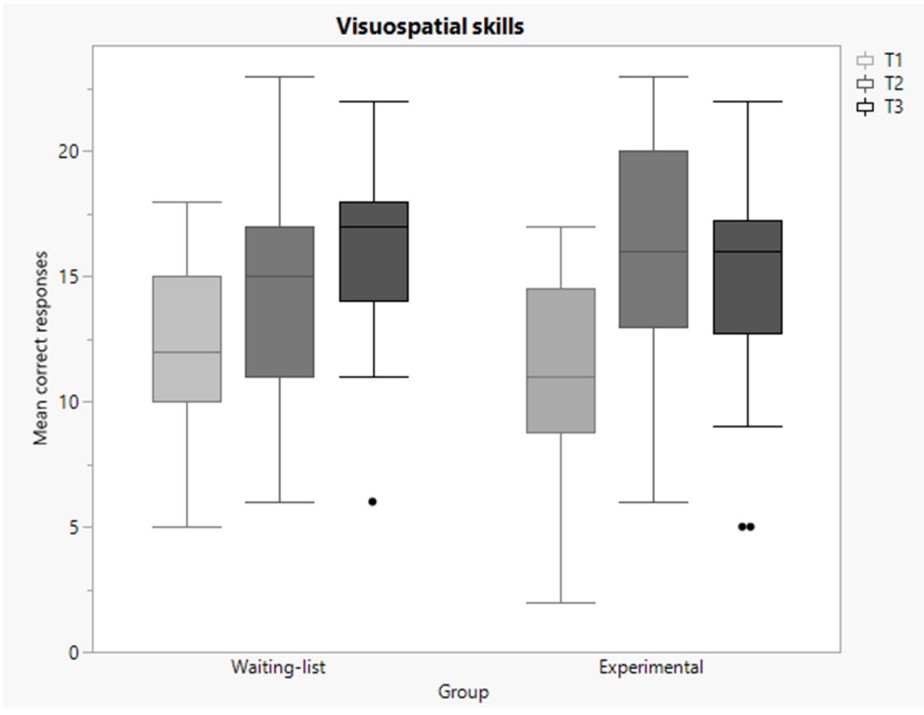

**Figure 16.** Groups' accuracy on visuo-spatial task at the pretest (T1), posttest (T2), and delayed posttest (T3).

## 4. Discussion

This study investigated the effects of a 7-week combined unplugged coding and ER intervention aimed at developing young children's (5-year-olds) computational thinking, response inhibition, planning, and visuo-spatial skills. A stepped wedge cluster randomized trial design was used to test the effects of the intervention, with the experimental and waiting list control group receiving the intervention at different times (the former between T1 and T2; the latter between T2 and T3). Results suggested that, compared to a waiting list group, children in the intervention unplugged coding and ER group significantly improved their coding abilities and visuo-spatial skills between T1 and T2. With the coding training received between T2 and T3, the waiting list control group also improved in coding, showing at T3 levels of performance equivalent to those of the experimental group. These findings confirm previous evidence of near-transfer effects of ER interventions on coding skills from an early age [28]. In addition, our findings also suggest far-transfer effects of the intervention, on visuo-spatial skills, and, partly, on planning abilities. Conversely, only response inhibition seemed to be not influenced by the unplugged-ER training, confirming previous evidence from a meta-analysis that highlights the resistance of response inhibition to far-transfer effects [73].

### 4.1. Training Effects on CT Skills

Our first research objective was to determine the extent to which preschoolers' computational thinking skills could be improved by an instructional program combining unplugged coding sessions and ER activities. Compared to the waiting list group, children in the experimental group demonstrated improvements in coding skills, with a large effect size (r = 0.87), and at T3, after the training, the waiting list control group also improved significantly in coding, with a moderate-to-large effect size (r = 0.71). It is important to remark that coding abilities were assessed by plugged coding tasks, and thus children's improvements in coding reflected children's ability to transfer the coding skills trained during tangible (unplugged and ER) intervention in which agents were physical objects or other children in a physical environment to a dimension where coding problems were plugged and presented in a virtual world (on screen), and thus more abstract and did not allow to concretely experience in a physical setting the program outcomes. Notably, children were never trained for the virtual coding test, as teachers were unaware of the specific requirements of our computational thinking assessment. For these reasons, we interpret our findings as evidence that the intervention effectively favored generalization and transfer of computational thinking skills. These results confirm those of other recent studies on tangible interventions in preschool [28,29,31], which showed a general effectiveness of ER training [28] and unplugged coding [31] in developing early children's coding skills. Fu and colleagues (2023) found, for instance, a positive near-transfer effect of ~12 h ER activities on the algorithmic skills of 42 preschoolers at the age 5–6 years with a slightly smaller effect size (d = 0.77) than we found. The consistency of the findings suggests that these types of tangible coding may be particularly suitable for preschoolers to learn computational thinking processes. In the present study, we introduced children to coding through tangible activities. Others have recommended starting with concrete representations involving unplugged [13], hands-on practices that allow children to physically move things around instead of having to mentally simulate every state of the execution of a program. At this age level, learning takes place essentially based on sensorimotor experience. Starting from Marinus et al.'s [66] and Critten et al.'s [23] studies on the feasibility of teaching computational thinking to preschoolers in a tangible environment, we structured the activities aimed at fostering generalization. This was carried out by introducing the fundamental computational thinking concepts first (e.g., sequences and algorithms) through unplugged coding activities, which were those closer to the child's everyday experience, and gradually shifting to ER. We conjectured that this would allow children to consolidate the new computational thinking concepts gradually before experiencing programmable robots. Robot programming was introduced as an extension and generalization of the unplugged coding

experience. We also reasoned about the additional issue in using robotics with young children, since programming a robot also means taking the robot's perspective to understand which instruction to give or which program steps are needed. Previous research [23,66] highlighted that perspective taking can be cognitively difficult for younger children, and this may limit the child's experience of ER, it being particularly difficult for preschoolers to understand directions from the robot's point of view. As it will be discussed below, our combined unplugged and ER training, however, seemed effective in training children's perspective taking, as demonstrated by the posttest performance of the groups, which after the training were able to solve plugged coding tasks that greatly relied on perspective taking, and also performed better on visual rotation tasks. Improvements in CT skills were not specific to a set-task of highly familiar stimuli, as computational thinking skills were tested on untrained coding tasks.

### 4.2. Cognitive Abilities

Our second research goal was to investigate the extent to which a combined unplugged and ER intervention would improve children's visuo-spatial skills and EFs. The results revealed that the children exposed to the coding intervention between T1 and T2 improved significantly in visuo-spatial tasks with a moderate-to-large effect size (r = 0.69). Conversely, the waiting list group also did not improve in their visuo-spatial abilities at T3. It is possible that the limited effect of the training for this second group could be related to the period in which the last posttest assessment was carried out; the intervention ended at the completion of the school year when children were very tired and fatigued. We conjecture that this condition might have influenced the children's posttest performance (T3). The experimental group, in contrast, received the coding intervention between January and March and carried out the posttest (T2) in April. The positive effect of coding on these children's visuo-spatial skills has been observed earlier [53,62], and our findings provide further confirmatory evidence on this theme. Importantly, in this exploratory study we assessed complex visuo-spatial skills, such as mental rotation and spatial relations. The study of Brainin et al. [62] assessed these complex skills too and found that the improvements in complex visuo-spatial abilities observed for the active control group (trained with unplugged coding) were inferior to those shown by the experimental group trained with ER. Our results provide the first empirical evidence that learning coding in a combined unplugged and ER environment positively affects also these complex visuo-spatial abilities.

Although the focus of our coding intervention was primarily aimed at developing computational thinking skills, we also explored the transfer effects to two EFs (response inhibition and planning) that had resulted in sensitive-to-coding interventions in prior studies [2,52,53]. Preschool years are a critical period for the development of brain regions that subserve EFs abilities and, as such, might constitute a crucial developmental window to target the malleability of these EFs [18]. Previous research found a positive far-transfer effect of computational thinking on response inhibition and working memory in first graders who had followed 10-week ER laboratories [53]. Response inhibition was positively affected by ER, with a moderate-to-large effect size (d = 0.69). Also, Canbeldek and Isikoglu [59] showed the positive effects of combined ER, unplugged, and plugged coding training on improving eighty preschoolers' self-regulation skills. Conversely, and in line with other research [58], in our study we did not find a significant effect on response inhibition skill. Our results are also consistent with the results of Kassai and colleagues' recent meta-analysis [73] about the near and far-transfer effects on children's EFs skills. In fact, also this meta-analysis reports insignificant far-transfer effects of EFs intervention on inhibitory control, working memory, or flexibility. It showed that while performance on the EFs components that are trained significantly improves, these improvements do not easily transfer to the untrained components [73]. This suggests that achieving a far-transfer effect on EFs at this early age requires interventions that target exactly the specific component of EFs skills to be trained. This observation is also in line with the results of Di Lieto and colleagues [53]. Their ER intervention aimed to enhance EFs through 16 sessions that were

explicitly planned and structured to improve specific EFs. The authors found significant effects of the ER training on visuo-spatial working memory and inhibition processes. In addition to targeting specific EFs, the overall duration of the training was longer than in our trial: 20 ER sessions of 60 min each. It may be that our combined unplugged coding and ER intervention, which was primarily aimed at teaching basic computational thinking concepts, was less adept at scaffolding EFs for 4–5-year-old preschoolers. Past studies demonstrated that learning coding can be effective both for the development of coding skills and EFs in six-year-old children, at the beginning of primary school [2,52]. At this phase of cognitive development, one year difference is significant and can explain differences in outcomes. Moreover, as noted, the duration of our training could have been insufficient to consolidate children's cognitive gains. In our 14-session training, we devoted 3 sessions to activities aimed at developing directional skills (differentiating left from right) and reproducing sequences of colors or pictorial symbols and 5 sessions to unplugged coding, which were not directly aimed at training EFs. One session focused on the familiarization with the *BeeBot* robots, and 5 sessions were focused on ER. Following prior research [23,74], in which preschoolers were first exposed to unplugged coding and then to *BeeBot* and *Cubetto* robots, we hypothesized that generalization would be facilitated by shifting from unplugged to ER activities. It may instead be the case that shifting between tools and types of activity in this short time was insufficient for consolidating EFs, and particularly inhibition skills. The lack of significant effects of the training on children's EF might also be related to the insufficient power of the study. Although significant effects were found for coding and visuo-spatial skills, the effects only approached significance for planning. It could be that inter-individual variability in this EF was still large at this age, as the literature suggests [75].

## 5. Limitations and Future Directions

The current study has some distinct limitations, which should be acknowledged.

The main limitation is that this study cannot determine which tool (unplugged coding or ER) is most effective for fostering the development of computational thinking and enhancing cognitive abilities such as response inhibition, planning, and visuo-spatial skills. The present study tested the effectiveness of a combined unplugged coding and ER intervention and not the intervention modalities independently. Although the findings of this exploratory study highlight the efficacy of these activities combined, it cannot provide information on the unique contribution of each intervention component (unplugged coding and ER). That is, based on our study we cannot conclude whether the intervention was effective because of its integrated nature or because of the effectiveness of just one of the two intervention components (e.g., unplugged coding or ER).

As in previous studies [2,76], we assessed response inhibition and planning by standard EFs tasks. We have used one task for each cognitive ability considered. We could have used more than one task for each executive function to increase the reliability of EFs assessment. However, longer assessment sessions could have been tiring for preschoolers with impact on their performance. The final phase of the present exploratory study took place at the end of the school year, a moment in which children might have been tired and cognitively fatigued. Performance, especially, during the follow up, might have been affected by the length of the assessment or repeated assessment sessions.

Lastly, it is worth noting that the sample size in this study was relatively small, and the children involved in our research project were all from middle-to-high socio-economic status, and therefore they were not representative of the population of preschoolers in Italy.

These considerations suggest the need for further studies, (1) to extend these findings to a larger and stratified sample, (2) to test the effects of intervention duration and dosage, and (3) to compare the effectiveness of the different intervention components like unplugged coding, ER, and their combination. Due to the lack of true experimental intervention studies, little is still known about which coding tools or programs (e.g., plugged or unplugged coding, unplugged coding or ER) can be most effective in developing coding ability and boosting children's cognitive skills. With other recent empirical studies, this

trial represents a contribution in this direction, but further research is needed to fill these gaps, and compare different coding tool, such as (1) combined unplugged coding and ER; (2) unplugged coding; (3) and ER interventions with younger learners.

Another future direction would be to verify the effectiveness of tangible coding on the development of coding skills and cognitive functions in children from disadvantaged socio-economic level.

## 6. Conclusions

Although interest in coding has increased much in the last few years [21,77], research focused on the cognitive effects of these activities is limited. Although recent studies suggest a positive effect of coding on the development of first graders' EFs, no studies have examined these effects in preschoolers as yet. With other recent studies, the present exploratory study is one of the first steps in bridging this gap.

The randomized controlled trial reported in this paper is the first finding about the effectiveness of a combined unplugged and ER training on the development of preschoolers' coding and cognitive abilities. The study shows that learning computational thinking concepts by tangible coding during the last year of preschool not only significantly improves children's abilities to solve coding problems (near-transfer effect), but it may also have some far-transfer effects on cognitive functions, such as visuo-spatial skills. In the current study, the benefits of coding learning have been observed in the preschool period, which has been shown to be a particularly sensitive time window for cognitive development.

As we highlighted in the introduction of this study, evidence about coding efficacy is needed to orient instructional decisions regarding which tools and programs are best to introduce young children to coding activities in preschool. Although the teaching of CT is compulsory, we still know very little about what programs work best and, consequently, sound recommendations for instructional practice are also lacking. The present study contributes to this field, designing and testing an instructional program for teaching coding from an early age.

**Author Contributions:** Conceptualization, C.M.; methodology, C.M.; software, G.P.; validation, C.M., G.P. and C.P.; formal analysis, C.M., B.A. and L.R.; investigation, C.M., G.P. and C.P.; data curation, C.M., G.P. and C.P.; writing—original draft preparation, C.M.; writing—review and editing, B.A.; review, T.V.; supervision, B.A. and T.V.; project administration, C.M. All authors have read and agreed to the published version of the manuscript.

**Funding:** This research received no external funding.

**Institutional Review Board Statement:** This study was conducted in accordance with the Declaration of Helsinki, approved by the Institutional Review Board at the University of Padua and follows EU GDPR 679/2016 Regulations for the Protection of Human Subjects Requirements [protocol number: 5113].

**Informed Consent Statement:** Written informed consent was obtained from all parents of participants involved in the study.

**Data Availability Statement:** Data will be available at https://osf.io/dtkm2/; accessed on 25 August 2023.

**Acknowledgments:** The authors acknowledge the support by the teachers from the San Pio X school at Padova, Italy. We wish to thank Valentina Francato for kindly helping with administrative and bureaucratic issues, as well as Alessia Vignola and Elisa Petrin for their great support throughout this study and for their contribution to the design some of the training activities. We also are grateful to Benedetta Pozzi for her special contribution in drawing the pictures for the CoThi platform, which the children loved and enjoyed so much. We also wish to thank Elena Martin, Simona Russo, Asya Petoukhov, and Carolina Braghenti for helping with the data collection. Finally, we would like to thank the children and their families for contributing to this study with great joy, interest, and motivation.

**Conflicts of Interest:** The authors declare no conflict of interest.

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
