# Peer review of "Combined Unplugged and Educational Robotics Training to Promote Computational Thinking and Cognitive Abilities in Preschoolers"

_education, doi:10.3390/educsci13090858_

Round 1

Reviewer 1 Report

Congratulations for the paper, it is very interesting and the quality and the structure of it is, in my opinion, very good.

I would suggest to add a section of conclusions and future lines, after the discussion. In fact parts of the discussion are conclusiones. For example the first paragraph and the last paragraph of the discussion could be the conclusions; maybe you have to change a little, but think about it, in order to have both sections. I believe that this could improve the paper.

Reviewer 2 Report

Overall, this paper is interesting and timely. I do have several comments:

Abstract

Comment 1

Lines 6-9 – This sentence is is a bit confusing to me. I would suggest making it clearer. It seems that unplugged coding and ER are barriers, but that is what you are testing.

Comment 2

Lines 23-25 – I would suggest including the stats that support these findings here in the abstract.

Comment 3

I would suggest to end the abstract with a sentence interpreting the results.

Introduction

Comment 4

Line 54-55 – What does “extra-European” mean?

Comment 5

Lines 88-94 – It seems that a citation is missing.

Comment 6

Line 86 – Explain this further, is there an exam required from preschool to primary school?

Comment 7

Line 131-133 – Add citation example (e.g., citation) since you are saying that other studies have demonstrated this.

Comment 8

Line 292 – You mentioned that this is a pilot study, I would suggest mentioning this in the abstract and maybe include it in the title. Or mention it earlier in the introduction when you talk about the study.

Comment 9

Lines 294-298 – Your research questions are “yes” or “no” questions. The answers to these questions are more complex, so I would suggest to rephrasing it a bit to maybe something like: “How can…”

Methods

Comment 10

For the procedure of the intervention, it looks like the intervention was given to children in groups. How did you ensure that all the children participated equally in the activities? An explanation of this needs to be added.

Comment 11

Line 597 – for the coding section, it is not clear to me if the assessment was done individually for each child or as a group. Please clarify if this measurement was done individually or in groups. Also, please clarify how this assessment was scored.

Comment 12

Where all the measurements done individually? If so please clarify.

Comment 13

You mentioned in the paper that you obtained informed consent from parents, but you did not mention if you got approval from the Institutional Review Board (IRB) or Ethics Department at your institution. I would suggest to also include this.

Discussion

Comment 14

Something that was not addressed that I think would be valuable to add in the discussion is the fact that your study tested the efficacy of unplugged coding + ER. But it would also have been good to see if one of these interventions alone would have been more effective than the other. Since you mentioned in the intro that unplugged coding alone or combined with ER has not been tested with preschoolers, so testing this here would have been interesting. For example, how do you know which, unplugged coding or ER had a greater influence on your results. You would not really know since you tested the combination of the interventions, not independently. Future research perhaps can test 1) unplugged coding + ER, 2) unplugged coding alone, 3) ER alone, 4) waitlist control.

Reviewer 3 Report

The work presented is interesting and provides valuable information for the field of study. The work is rigorous and easy to understand. However, it is believed that the work could be improved by following the following aspects:

- In the abstract it is stated that there are fifty-seven participants and then it says 47.

- The information in the theoretical framework is very long and it would be advisable to reduce it a little.

- Point 1.2 could be eliminated or reduced as the information on the study itself should be included in the methodology. 

- With regard to the evaluation instruments used, it is not stated what type of validity they have and it would be advisable to indicate this. 

- The discussion could be improved with more references.

- Indicate the limitations of the study.
